# Post-COVID-19 Anosmia and Therapies: Stay Tuned for New Drugs to Sniff Out

**DOI:** 10.3390/diseases11020079

**Published:** 2023-05-27

**Authors:** Gabriele Riccardi, Giovanni Francesco Niccolini, Mario Giuseppe Bellizzi, Marco Fiore, Antonio Minni, Christian Barbato

**Affiliations:** 1Department of Sense Organs (DOS), Sapienza University of Rome, Viale del Policlinico 155, 00161 Roma, Italy; gabriele.riccardi@uniroma1.it (G.R.); giovannifrancesco.niccolini@uniroma1.it (G.F.N.); mariogiuseppe.bellizzi@uniroma1.it (M.G.B.); 2Institute of Biochemistry and Cell Biology (IBBC), National Research Council (CNR), Unit of Translational Biomolecular Medicine, Department of Sense Organs (DOS), Sapienza University of Rome, Viale del Policlinico 155, 00161 Roma, Italy; marco.fiore@cnr.it; 3Division of Otolaryngology-Head and Neck Surgery, Ospedale San Camillo de Lellis, ASL Rieti-Sapienza University, Viale Kennedy, 02100 Rieti, Italy

**Keywords:** anosmia, smell, post-COVID-19, olfactory impairment, therapy, clinical trials

## Abstract

**Background:** Anosmia is defined as the complete absence of olfactory function, which can be caused by a variety of causes, with upper respiratory tract infections being among the most frequent causes. Anosmia due to SARS-CoV-2 infection has attracted attention given its main role in symptomatology and the social impact of the pandemic. **Methods:** We conducted systematic research in a clinicaltrials.gov database to evaluate all active clinical trials worldwide regarding drug therapies in adult patients for anosmia following SARS-CoV-2 infection with the intention of identifying the nearby prospects to treat Anosmia. We use the following search terms: “Anosmia” AND “COVID-19” OR “SARS-CoV-2” OR “2019 novel coronavirus”. **Results:** We found 18 active clinical trials that met our criteria: one phase 1, one phase 1–2, five phases 2, two phases 2–3, three phases 3, and six phases 4 studies were identified. The drug therapies that appear more effective and promising are PEA-LUT and Cerebrolysin. The other interesting drugs are 13-cis-retinoic acid plus aerosolized Vitamin D, dexamethasone, and corticosteroid nasal irrigation. **Conclusions:** COVID-19 has allowed us to highlight how much anosmia is an important and debilitating symptom for patients and, above all, to direct research to find a therapy aimed at curing the symptom, whether it derives from SARS-CoV-2 infection or other infections of the upper airways. Some of these therapies are very promising and are almost at the end of experimentation. They also provide hope in this field, which not addressed until recently.

## 1. Introduction

The COVID-19 (coronavirus disease 19) outbreak has rapidly spread around the world since December 2019, resulting in more than 700 million confirmed cases and 6.5 million reported deaths from the virus. Symptoms reported by SARS-CoV-2 (severe acute respiratory syndrome coronavirus 2) infected patients include fever, dyspnoea, olfactive impairment, anosmia, dysgeusia, myalgia, malaise, headache, sore throat, rhinorrhoea, diarrhoea, nausea/vomiting, nasal and conjunctival congestion, and neurological symptoms defined as neurocovid [1]. The prevalence of olfactory dysfunction in patients with COVID-19 is estimated at 52.73% [2]. The work of Kaye et al. reported anosmia in nearly 73% of patients and showed that the symptom preceded the diagnosis of COVID-19. Another interesting finding is that anosmia was the initial symptom in over 26.6% of these patients. Although most COVID-19-related olfactory dysfunctions seem to recover after a short time, i.e., within 30 days, some patients have reported long-term anosmia [3]. From an anatomical–physiological point of view, the nasal cavity is lined by two different types of epithelia: respiratory and olfactory epithelium. The respiratory epithelium is pseudostratified columnar epithelium consisting of ciliated cells, secreting goblet cells, and basal cells [4]. It represents a major part of the mucous membrane of the nasal cavities. The olfactory epithelium is responsible for the perception of olfactory sensation and consists of at least five cell types: sustentacular cells, olfactory sensory neurons (OSNs), microvillar cells, basal cells, and Bowman’s glands. OSNs express, on cilia membranes, G-protein-coupled receptors for the detection of odorant molecules by odorant binding-induced activation. This activation results in the stimulation of adenylate cyclase and the formation of cyclic adenosine monophosphate, which leads to the opening of cyclic nucleotide-gated channels for the influx of sodium and calcium ions and subsequent calcium-activated efflux of chloride ions at the olfactory cilia, creating a receptor potential for triggering action potentials at the soma. They are bipolar neurons and therefore form synapses with the olfactory bulb through their axons which project into the nasal lamina propria and the cribriform plate (Figure 1) [4]. Each olfactory sensory neuron has a single type of olfactory receptor, the signals of which are convergently transmitted to one or two glomeruli and thereby connecting tufted and mitral cells in the olfactory bulb. These cellular axons are assigned to various olfactory areas in the central nervous system [4].

The causes of anosmia can be various. The most common ones are sinus disease, postinfectious disorder, and post-traumatic stress disorder. Gradual onset and difficulty remembering a triggering event could suggest a neurodegenerative disease or age-related disorder. On the other hand, acute onset, or the presence of exacerbations on chronic pictures suggests sinus disease as the cause [5,6]. An otorhinolaryngology examination for olfactory impairments should include nasal endoscopy to visualize the olfactory cleft and to rule out any nasal disease [7]. Specifically, care should be taken to detect septal deviations, tumors, and signs of acute or chronic sinus disease, such as discharge, crusting, and polyps. Testing of olfactory function is mandatory because subjective assessments of olfactory function are not always reliable [8,9]. Smell testing is usually performed with the University of Pennsylvania Smell Identification Test (UPSIT) or Sniffin’ Sticks test battery (threshold, discrimination, and identification (TDI) score) [10,11]. If deemed necessary, diagnostic imaging tests such as computed tomography (CT scan) or magnetic resonance imaging (MRI) can be performed. Imaging studies can be useful not only to highlight potential causes but also because the volume of the olfactory bulb can be used to predict the prognosis of olfactory dysfunction [12]. The therapies that can be used are limited, as they are represented by oral and intranasal corticosteroids and by olfactory training with Zinc-Gluconate [13]. For this reason, our review aims to highlight the active clinical trials on the medical therapy of anosmia, focusing on those in an advanced and more promising stage. Therefore, our intent is to explore which drugs we can focus on in the coming years as ENT specialists in the management of olfactory impairment.

## 2. Materials and Methods

The clinicaltrials.gov database was interrogated as a source for the present investigation. Clinicaltrials.gov is a web-based resource maintained by the National Institute of Health that updates information on public and privately supported clinical studies. All clinical investigations of Food and Drug Administration (FDA)-regulated drugs or medical devices are registered on this database. Furthermore, it also represents a database for many clinical trial protocols conducted around the world. It was interrogated on 1 December 2022, using the following search terms: “Anosmia” AND “COVID-19” OR “SARS-CoV-2” OR “2019 novel coronavirus”. In clinicaltrials.gov, the advanced search function was used to restrict interventional studies and the drug protocol. Three reviewers (G.R., G.F.N., and M.G.B.) screened the identified protocols to remove duplicates and verify the inclusion criteria: (1) targeting anosmia after SARSCoV-2 infection and/or clinical syndromes associated with COVID-19 and (2) testing the efficacy and/or safety/tolerability of pharmacological interventions. Disagreements in the selection were solved involving an additional reviewer (C.B.). Data Extraction: Data were abstracted by (G.R., G.F.N., and M.G.B.) from the selected protocols: NCT (the unique identification code assigned by clinicaltrials.gov); study phase; allocation and masking procedures; tested compound(s); tool of administration; mechanism of action; primary outcome measure(s); expected primary completion date; expected number and age of participants.

## 3. Results

A total of 23 clinical trials were identified. Five protocols were excluded because they did not target anosmia after COVID-19. Accordingly, 18 studies were ultimately retained. Selected clinical trials using new drugs are commonly classified into four phases. The selected trials encompassed more than one phase (e.g., combined phases 1–2). Overall, one phase 1, one phase 1–2, five phase 2, two phase 2–3, three phase 3, and six phase 4 studies were identified. Their detailed characteristics are presented in Table 1. Most trials were conducted in the US (*n* = 4) and in Egypt (*n* = 4), followed by Canada (*n* = 2), Italy (*n* = 1), China (*n* = 1), Iraq (*n* = 1), Mexico (*n* = 1), France (*n* = 1), Belgium (*n* = 1), Ukraine (*n* = 1), and Hong Kong (*n* = 1). Protocols had varying durations and were expected to be completed (in terms of primary completion) between September 2020 and December 2022. Fourteen trials had a randomized design on a parallel assignment of participants. Three trials were built with a single group assignment and one with no randomized design on a parallel assignment (Table 1 and Figure 2).

### 3.1. Phase 1 Clinical Trials

In only one phase 1 clinical study, the efficacy of local steroids, through the intranasal administration of betamethasone, was estimated in post-COVID-19 anosmic patients [28].

### 3.2. Phase 2 Clinical Trials

The study group of Mount Sinai Hospital thought to investigate the efficacy of omega-3 fatty acid supplements in post-COVID-19 patients with anosmia. Omega-3 polyunsaturated fat supplementation has emerged as a valuable pharmacotherapy for olfactory dysfunction. In the mouse model, a diet without omega-3 fatty acids led to the onset of olfactory dysfunction; conversely, mice receiving omega-3 fatty acids showed olfactory recovery following peripheral nerve injury, suggesting a neuroprotective effect mediated by antioxidant and anti-inflammatory pathways. A wide cross-sectional study evidenced that older adults with higher dietary fat intake had a lower incidence of olfactory impairment [29]. In this direction, patients without sino-nasal disease receiving postoperative omega-3 fatty acid supplementation after endoscopic endonasal skull base surgery showed a significantly greater rate of normal olfactory function [19]. Patients were randomized into a control group and received treatment with omega-3-fatty acid supplementation (1000 mg of omega-3 fatty acid blend including 683 mg Eicosapentaenoic Acid and 252 mg Docosahexaenoic Acid) twice daily for 6 weeks. The smell was tested with the brief smell identification test (BSIT), with a full range score from 0 to 12, where a higher score indicates better olfactory at the beginning and end of administration [29].

Since post-COVID-19 anosmia etiopathogenesis is unknown, it is suggested that sensory axonal regeneration and olfactory signaling might rely on elevated levels of cAMP and cGMP. Piccirillo et al. proposed using theophylline, a molecule involved in this pathway [19]. They performed a phase II treatment trial, and patients were allocated 1:1 to receive either intranasal theophylline irrigation for six weeks: 400 mg theophylline capsule diluted in 240 mL isotonic nasal saline lavage, twice daily, or a placebo: 500 mg lactose capsule diluted in 240 mL isotonic nasal saline lavage, twice daily. Various tests and surveys were utilized to capture changes in smell ability in the two groups of patients to compare 6 weeks post-intervention from baseline, e.g., the Clinical Global Impression Scale Questionnaire for Olfactory Dysfunction (QOD) and Olfactory Dysfunction Outcomes Rating (ODOR) [30]. Another study from the same group investigated the efficacy of oral gabapentin, an anti-epileptic drug used for nerve pain, in olfactory improvement following COVID-19-associated olfactory dysfunction [31]. In this clinical trial, the patients were divided into two groups: (1) in the first group, the drug was administered over a maximum of 14 weeks with up to four weeks titrating up, eight weeks maintaining the highest tolerable dose, and up to two weeks tapering down; (2) in the second group, placebo gelatin capsules that look, smell, and taste like gabapentin was administered to the placebo arm, with a similar dosage and method to preserve double-blinding of the study. The tests and scales used to monitor the effectiveness and progress of the trial were implemented with the addition of NASAL-7. The test contained seven household items, with each item scored as 0 (cannot smell), 1 (smells less strong/different than normal), and 2 (smells normal), with a potential score in the range of 0–14 [20]. This study evidenced that the clinical benefit of theophylline nasal irrigations in participants with COVID-19-related OD is inconclusive, but other studies could investigate the efficacy of this treatment more fully [20].

At the Montreal Heart Institute, a trial with the aim to determine the effects of short-term treatment with hesperidin, a medicinal plant with antiviral action, on COVID-19 symptoms was completed. The 216 participants were divided into two groups: the active and the placebo comparator. Patients received an investigational drug (two capsules of 500 mg of hesperidin) at the same time in the evening, at bedtime, with water. The Placebo was administered with a similar dosage to maintain the blinding of the study. Treatment effects were observed through a symptom diary by participants throughout the study and by taking the oral temperature daily [32].

Hung’s group from Hong Kong led a clinical trial to investigate the safety and therapeutic efficacy of oral vitamin A in combination with intense aromatic chemosensory smell training (ST) by pulse aromatic stimulation. The 25 patients were divided into three groups: (1) a 14-day course of daily oral vit A 7500 µg RAE (retinol activity equivalents) in combination with ST three times per day for 4 weeks; (2) ST three times per day for 4 weeks alone; (3) observation only. After a period of 4 weeks, the patients were assessed using the subjective olfactory assessment (SNOT-22), objective olfactory assessment by the butanol threshold test (BTT), and objective olfactory assessment by the smell identification test (SIT) [33]. The results of this study are not available yet.

### 3.3. Phase 1–2 Clinical Trials

In the only clinical trial in phases 1–2, a single cohort study was utilized to generate pilot data on the efficacy and safety of sequential stellate ganglion blocks for the treatment of COVID-19-induced olfactory dysfunction and other long COVID symptoms [34]. This hypothesis was suggested because other symptoms, such as chronic dyspnoea, impaired memory capacity, significant fatigue in positive sympathetic feedback loops, and dysautonomia, were observed. This study proposed to block stellate ganglion and hyper-sympathetic activation by inhibiting sympathetic neuronal firing and resetting the balance of the autonomic nervous system. After ultrasound guidance identification, 1% lidocaine or 1% mepivacaine was injected near the ganglion stellate. They assessed the efficacy with UPSIT, C-GIT, and ODOR at baseline, after 1 week, and after 1 month. The recruitment status was completed, but no study results were posted on ClinicalTrials.gov.

### 3.4. Phase 2–3 Clinical Trials

In Egypt, South Valley University started a clinical trial to assess the efficacy of intranasal ivermectin administration in patients with anosmia [18]. This drug is an approved and effective antiparasitic with antiviral effects in vitro. The authors recruited patients with post-COVID-19 anosmia, divided them into two groups, and administered ivermectin nanosuspension and saline nasal sprays to the treatment and placebo groups, respectively. The ability to regain smell after 14 days of starting therapy was evaluated. Horoi et al. investigated the usefulness and safety of platelet-rich plasma (PRP) injection in 56 patients with COVID-19 and chronic olfactory dysfunction; in patients with non-COVID-19 anosmia, they reported encouraging results [25]. One month after the PRP injection in the olfactory cleft, the mean TDI scores were significantly improved by 6.7 points (from 21.3 ± 7.4 to 28.0 ± 5.0), in contrast to 0.5 points (from 24.5 ± 7.4 to 25.0 ± 7.7) in the control group. These results suggest, on the one hand, that the timing of treatment may be an important factor and, on the other hand, that PRP is a safe treatment because it has no adverse effects.

They divided the patients enrolled into an experimental group and a control group. The investigators compared the result of the PRP group to a control group that underwent simple olfactive training for one month with the sniffing stick test (TDI score) and a linker scale from 0 (none) to 3 (strong).

### 3.5. Phase 3 Clinical Trials

The clinical trial at Benha University evaluated the role and effectiveness of topical corticosteroid nasal spray, the mometasone furoate, in patients with anosmia after COVID-19 infection [35]. Group I received topical corticosteroid nasal spray (mometasone furoate) in addition to olfactory training, and Group II received only olfactory training. The assessment of smell was conducted using familiar substances with a distinctive odor, such as coffee, mint, and garlic. The patients reported the degree of anosmia subjectively with a score on a scale from 0 to 10. This process was carried out initially, after 1 week, 2 weeks, and 3 weeks for all patients. Mometasone furoate nasal spray, used as a topical corticosteroid in treating post-COVID-19 anosmia, showed no superior benefits over olfactory training [36]. Another clinical trial evaluated therapy with nasal corticosteroid administration in the efficacy of local budesonide in the management of persistent hyposmia in COVID-19 patients [37]. In the experimental group, nasal irrigation with budesonide and physiological saline was administered twice a day for 30 days, in addition to olfactory rehabilitation, in the morning and in the evening. In the control group, only nasal irrigation with physiological saline was administered, morning and evening, for 30 days, in addition to olfactory re-education. They assessed the results with ODORATEST after drug administration therapy. To date, no official results have been released from this study.

The last Phase 3 study to be published aimed to quantify improvement in the olfactory function of 27 patients after COVID-19 infection after three administrations of intranasal insulin during a four-week period with the evaluation of TDI scores by Sniffin’ Sticks [21]. This study suggested a new management method for hyposmia patients before the pandemic period without unwanted effects [21]. Recently, intranasal insulin fast-dissolving films for the treatment of anosmia in patients post-COVID-19 infection were investigated [38]. In total, 40 IU of NPH insulin was placed on the nasal roof between the nasal septum and the middle meatus. During the fourth and last visit, olfaction was re-evaluated using the measures previously described. Despite the limited number of patients, this study showed that intranasal insulin (40 IU) administration was able to increase odor identification, suggesting that insulin-dissolving films could be a promising approach to anosmia treatment [38].

### 3.6. Phase 4 Clinical Trials

The first study in Phase 4 was a multicentric study that addressed comparing the efficacy of neuro-protective and anti-inflammatory agents palmitoylethanolamide (PEA) and Luteolin (LUT) with control subjects (olfactory training). The cohort of patients under observation showed persistent smell disorders after resolution from COVID-19 and negative swabs for 4 months [39]. Patients were evaluated at baseline with Sniffin’ Sticks before initiating scent training and/or supplemental treatment (T0). One group received daily olfactory training, and the other group additionally received a daily oral tablet that contained PEA 700 mg plus Luteolin 70 mg (one dose of PEA-LUT only or two doses). Assessment of olfactory function was conducted at 30, 60, 90, and 120 days. The results show that 92% of patients in the intervention group improved versus 42% of controls, suggesting that PEA-LUT with olfactory training resulted in greater recovery of smell than olfactory training alone [16,40].

Another study hypothesized that Cerebrolysin, a neurotrophic and neuroprotective drug, can be used to treat patients with persistent post-COVID anosmia or ageusia and promote functional recovery of smell and taste deficits [19]. Previous preclinical and clinical studies have shown that treatment with neurotrophic polypeptides can promote neurological recovery for many neurodegenerative and chronic nervous system diseases.

Cerebrolysin is a mixture of porcine-derived neuropeptides and free amino acids, including nerve growth factor (NGF), brain-derived neurotrophic factor (BDNF), ciliary neurotrophic factor (CNTF), enkephalins, orexin, and P21 [41,42], approved for use as a treatment for dementia [43], stroke [44], and traumatic brain injury (TBI) [45]. They administered 5 mL of Cerebrolysin in an ampoule once daily through intramuscular injection five times per week for 8 weeks, after which the cycle was individually repeated according to the response of the patient to therapy for a maximum of 24 weeks. Unexpectedly, in this study, a control group was absent. Recovery and improvement of smell were evaluated with the Globas rating for smell (GRS) and the Globas rating for taste (GRT). These evaluations were carried out at the baseline, after 8, 12, 18, and 24 weeks of therapy, and it is still ongoing. There is experience with the use of the drug Imupret for the treatment of nasopharyngitis associated with other viral pathogens [46]. Popovych et al. compared Imupret with the standard of care in clinical trials because it is important in relation to the activation of non-specific immunity factors compared to COVID-19 infection. A potential application of Imupret for the therapy of nasopharyngitis associated with COVID-19 would allow the development of new therapeutic tools to counter this infection and suggest it into clinical practice. In this multicentric trial, they divided the patients into two groups: the first group was treated with Imupret, with a dosage of 25 drops six times for 14 days, and in the control group, the standard of care, as symptomatic therapy as needed, was implemented, with paracetamol and saline solutions in the nose four times a day for 14 days. The assessment was performed with the control and evaluation of symptoms, such as sore throat, fever, rhinorrhoea, cough, and nasal congestion.

Recent findings showed that COVID-19 binds directly to STRA6 receptors, leading to retinol depletion [47]. Aerosolized retinoic acid may have an effective role in treating post-COVID-19 anosmia through upregulating ACE2 and STRA6 and regenerating OSNs expressing olfactory receptors. Elkazzaz’s study group investigated the potential role of aerosolized retinoic acid in treating COVID-19 anosmia [48]. They divided the patients into three groups: (1) the first group of patients received a daily dose of 13-cis-retinoic acid aerosol with gradual increases in two doses from 0.2 mg/kg/day to 4 mg/kg/day for 3 weeks. In addition, this group received 600,000 units of cholecalciferol (vitamin D) intramuscular injection for two doses at week 0 and week 4. (2) The second, on the other hand, received a daily dose of all-trans retinoic acid aerosol in two doses, gradually increasing from 0.2 mg/kg/day to 4 mg/kg/day by inhalation for 3 weeks. Additionally, these patients received oral supplementation. (3) The third group of patients was administered the standard of care. The improvement in olfaction 3 weeks after beginning to take supplements was evaluated. This study is still ongoing.

In another trial in Phase 4, the authors wanted to assess the recovery of anosmia with early corticosteroid use. They evaluated the administration of dexamethasone 6 mg/day per os and 4 mg/day intravenously in two groups of 15 patients with community-acquired pneumonia. The study aimed to use dexamethasone as early laboratory evidence of high inflammatory markers compared to late dexamethasone administration upon the deterioration of cases. The time frame of 6 weeks to recover from anosmia was evaluated, and the study is running [22].

The last clinical trial planned to involve patients with anosmia onset immediately after an upper respiratory viral illness and assigned them to three distinct study arms [49]. Nasal irrigation was prescribed to all three groups. In addition, one group received a datasheet hand-out about post-viral anosmia with instructions to smell common household items (current care) and act as a control group. The second group received an essential oil retraining kit, whereas the third group received the same olfactory training kit and a prescription to use budesonide with nasal irrigations. Olfactory scores were tested at enrolment and at 3–6 months. The literature reported that adding budesonide irrigation to olfactory training significantly improved olfactory ability compared with olfactory training plus saline irrigation.

## 4. Discussion

From the recognition of active clinical trials aimed at olfactory dysfunction therapy, the most promising drugs seem to be PEA-LUT and Cerebrolysin (Figure 3). PEA is a bioactive lipid mediator with the same effects as endocannabinoids synthesized on demand within the cell, and we can find it in all tissues, including the brain [50]. PEA has a pro-homeostatic protective action against cell damage and is usually upregulated in pathological states. PEA pleiotropic effects include anti-inflammatory, analgesic, anticonvulsant, antimicrobial, antipyretic, anti-epileptic, immunomodulatory, and neuroprotective activities [51,52]. PEA has a tropism mainly at the nuclear peroxisome proliferator-activated receptor alpha (PPAR-α) and acts on the novel cannabinoid receptor, G protein-coupled receptor 55 (GPR55), and G protein-coupled receptor 119 (GPR119) [53]. Furthermore, it activates and desensitizes channels, contributing to a significant anti-nociceptive effect [54]. It does this through several mechanisms, including the entourage effect, by activating PPAR-α and potentially acting as an allosteric modulator [55]. PEA’s inhibition of mast cell (MC) activation also plays a role in local autacoid inflammation (ALIA) antagonism [56]. Hence, the binding of PEA to PPAR-α receptors of immune cells such as macrophages and MCs assigns a role in the reduced production of inflammatory signals and pain signals. It should also be emphasized that it also modulates the action of interleukins, thus also mediating inflammation [57]. PEA carries out its neuroprotective action by modifying and regulating the activation of MCs, microglia, and astrocytes. Specifically, it performs this action by improving microglial migration without promoting its activation but increasing resistance to infections without measurable pro-inflammatory effects. PEA is an immunomodulator instead of an immunosuppressant (Figure 4). PEA inhibition of pro-inflammatory cytokines most likely reveals its role in the prevention of cortical diffusion depression in preclinical models and its therapeutic effects in migraines [58]. The mechanisms mentioned above are explanatory of the protective effects of PEA for several neurological pathologies evidenced in ex vivo and mouse models of Alzheimer’s disease, Parkinson’s disease, ASD, stroke, and traumatic brain injury [59,60,61]. In these models, chronic administration of PEA (10–100 mg/kg) attenuated gliosis and neuroinflammation, protecting against neuronal degeneration and apoptosis, rescuing glutamate toxicity, inhibiting oxidative stress processes, and ameliorating behavioral and cognitive deficits [62].

To manage anosmia in this study, the researchers administered PEA with LUT [63]. It is a flavonoid (3,4,5,7-tetrahydroxy flavone) contained in plants, vegetables, medicinal herbs, and fruits. LUT exhibits multiple biological effects, such as anti-inflammatory, anti-allergic, and anti-tumor, and can function as both an antioxidant and a pro-oxidant in vitro and in vivo. The biological effects of LUT could be functionally related to each other and act synergistically (Figure 4). As previously mentioned, Cerebrolysin is a parental-administered, low-molecular-weight neuropeptide preparation obtained by standardized enzymatic proteolysis of porcine brain proteins and exhibits neuroprotective and neurotrophic properties such as those found naturally in neurotrophic growth factors [64,65] (Figure 5). Cerebrolysin has been approved primarily for the treatment of dementia [44], acute ischemic stroke, cognitive impairment, and traumatic brain injury [63]. In cerebral autosomal dominant arteriopathy with subcortical infarcts and leukoencephalopathy (CADASIL) with cognitive impairment, a mild protective effect of Cerebrolysin against oxidative stress was observed [64]. Moreover, the protective potential of Cerebrolysin for PTSD-induced short-term and long-term memory impairment was attributed to cerebral proteases preventing increased oxidative stress in the PTSD hippocampus [65]. Several studies are currently evaluating Cerebrolysin efficacy for other conditions, including cerebral palsy, vascular dementia, aneurysmal subarachnoid haemorrhage, and anosmia [66]. The main pharmacological effect of Cerebrolysin is the restoration of the surrounding and damaged nerve structures, which is deleterious to neurons, causing their degeneration, dysfunction, and death [67]. Few articles have demonstrated the safety and efficacy of this drug, but noteworthy is a meta-analysis that demonstrated that this blend of peptides can be used as a safe therapy for recovery from severe traumatic brain injury. Additionally, a randomized controlled trial of infants at risk for neurodevelopmental delay showed its benefits as an add-on therapy to reduce the incidence of movement and speech delay or dysfunction. The most frequently reported self-limiting adverse reactions due to Cerebrolysin are vertigo or light-headedness, headache, nausea, and increased sweating. Some authors also reported urinary tract infections and fever as adverse reactions, but these were related to the infusion treatment rather than Cerebrolysin [66].

Other drugs that have been tested in clinical trials reached the last stage and are represented by local and systemic corticosteroids, Retinoic acid, and Imupret [12,13,26,48]. Corticosteroids can also be effective in treating olfactory loss and are commonly used by clinicians without knowing the cause of anosmia. Several clinical studies have demonstrated the benefits of olfaction after systemic corticosteroids beyond what the spontaneous recovery expected [68]. Therefore, they represent an effective but certainly non-specific therapy and reduced to reducing only one, however important, of the causal aspects of anosmia, i.e., inflammation. Regarding the use of aerosolized retinoic acid, the authors suggested that STRA6 function was lost due to a blockade by the COVID-19 spike protein, which binds to it with high affinity, hijacking the signaling pathway, leading to disruption of retinoic acid synthesis and vitamin A deficiency. This is because Elkazzaz et al. indicated that the administrated retinoic acid would facilitate cellular uptake of retinol in immune cells through the impaired transport function of the STRA6 receptor by binding to the RBD of the SARS-CoV-2 spike protein, leading to an improvement in immune system homeostasis [48,69]. This may also be applicable to olfactory epithelial homeostasis because retinoic acid insufficiency in the olfactory epithelium causes cell failure in progenitor cells and, consequently, olfactory neuron differentiation is not maintained. It has been proven that the renewal of olfactory neurons is inhibited if retinoic acid synthesis fails in the olfactory epithelium.

In the case of SARS-CoV-2, the spike protein interacts with the ACE2 receptor on the surface of epithelial cells, e.g., in the oral cavity or respiratory tract. The extracts from Imupret were able to interfere with the binding between the S1 spike protein and the human ACE2 receptor. Consequently, it activated human innate immune defense by increasing the level of defensin HBD1, an immunomodulatory, and/or cathelicidin LL-37, a multifunctional peptide involved in infection and inflammation in the lungs. It was evident at low concentrations in vitro and in vivo [70]. Both drugs seem to be very interesting, but the data provided by the investigators are not sufficient to provide a clinical role to this therapy in patients with post-COVID-19 anosmia. In addition to medical therapies, as described above, it is important to mention olfactory training, which is effective in several olfactory impairments. The multiplicity of exposure to selected odors daily and weekly showed an important recovery valuable by Sniffin’ Sticks [71].

A criticism observed in all clinical trials analyzed in this report was identified between different olfactory methods used for testing the therapy efficacy. This exciting issue is beyond the scope of this review. However, the diversity of olfactory tests used to evaluate the efficacy of pharmacological treatments in different patient cohorts represents a limit to the interpretation of the concluded studies. In addition, it is important to note that this review was aimed at recognizing the active clinical trials on anosmia therapy in post-COVID patients, deposited on clinicaltrials.gov, as described above in the Method section, and not on other olfactory functions or impairment. In this regard, a recent study elucidated the used language in clinical and research studies aimed to define the terms and definitions of olfactory nomenclature [72]. The authors suggest that dysosmia included quantitative and qualitative olfactory dysfunction and, in turn, proposed a definition measurable with olfactory testing (anosmia and hyposmia) and patient-reported subjective rating (parosmia, phantosmia, and olfactory intolerance). In effect, in all the trials analyzed, these considerations regarding non-heterogeneous olfactory tests, and term assignments of olfactory quantitative and qualitative definitions, represent bias.

In addition, regarding treatment effectiveness in long COVID anosmics, clinical experience is limited. This problem is mainly because research on human olfactory dysfunctions, investigations on the olfactive trans-duction mechanism, the cellular and molecular profile of neuroepithelial cells, and cognitive and memory elaboration of the olfactory area are still developing. Too little time has passed since the end of the pandemic to already have data available on what is defined as a post-COVID syndrome [4]. More recently, an international consensus on the management of post-COVID-19 olfactory dysfunction suggested that olfactory training remains the recommended management protocol, and treatment with systemic corticosteroids is not recommended until new studies are completed [73].

We discussed several therapeutics employed in several anosmia clinical trials, and among active trials, PEA and Cerebrolysin represented promising molecules acting to reduce olfactory impairment (Figure 6). As an alternative to drug administration, other methods could also be used. Conversely, another promising approach to olfactory loss could arrive from the activation or replacement of stem cell populations inside the olfactory epithelium. More recently, a new mouse model of inducible hyposmia was generated, and purified tissue-specific stem cells were shown to engraft intranasally to produce olfactory neurons, resulting in the recovery of function [74] (Figure 6). In addition to genetic, electrophysiological, and behavioral assays performed in adult mice, the authors explored, in vitro, the mechanisms promoting stem cell expansion of adult olfactory basal progenitor cells competent for engraftment, paving the way for a potential translational therapeutic approach [75]. Previously, we suggested that the pandemic boosted the occurrence of studies on olfactory diseases, and great efforts are required by the scientific community to explore new therapeutic avenues [4,76]. Olfactory dysfunction therapy can obtain new opportunities from several research areas, such as biologic therapies, with dupilumab, which has shown remarkable improvement in smell loss for chronic rhinosinusitis nasal polyposis patients [77] (Figure 6). Biologics therapy for improvement in olfactory function is not planned in anosmia to date and implementing this alone or joined with other standard treatments could be a future perspective (Figure 6). On the other side, a major effort could be performed in the diagnosis of smell dysfunction in specific populations. Recently, it was evidenced that the omicron variant caused less olfactory dysfunction than previous variants of SARS-CoV-2 alpha or delta. The reported prevalence differs greatly between populations, probably due to genetic differences, as suggested by a genome-wide association study that connected a gene locus encoding an odorant-metabolizing enzyme, UDP glycosyltransferase, to the extent of COVID-19-related olfactory impairment [78]. In conclusion, the aim to obtain an efficacy treatment for anosmia in post-COVID-19 patients will be a result of a combination of pharmacological, genetic, and electrophysiological studies, observing that anosmia is often present in a neurocovid context, a complex and articulated syndrome, where an alliance between researchers and physicians is necessary [79].

## Figures and Tables

**Figure 1 diseases-11-00079-f001:**
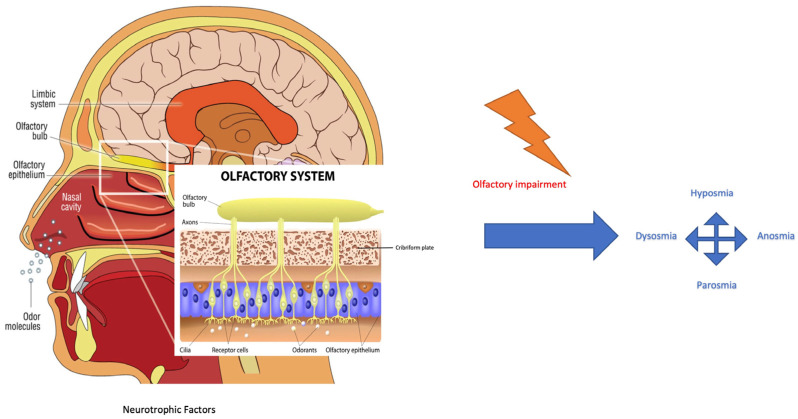
Odorant signal transmission and the human olfactory system. Focus is on the olfactory system and neuroepithelium, composed of olfactory sensory neurons that, through the lamina cribrosa plate, establish a synaptic connection with the mitral and tufted cells of the olfactory bulb. The axons of the monosynaptic mitral and tufted cells that make up the olfactory tract bifurcate at the terminus, fornix, or olfactory cortices. Primarily, the limbic system with the pyriform cortex, amygdala, and entorhinal cortex is involved [4]. Olfactory impairment induces modifications to smell, as reported in the text. (Modified from medical illustration by Patrick J. Lynch).

**Figure 2 diseases-11-00079-f002:**
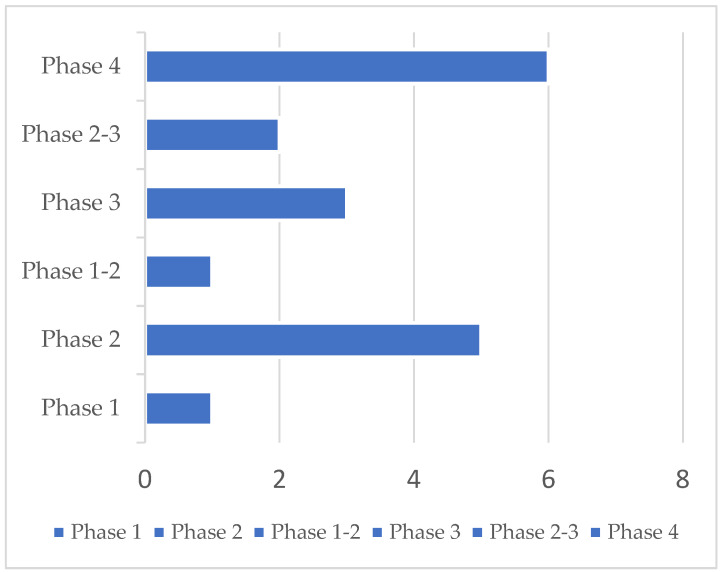
23 clinical trials were identified, 5 were excluded, and 18 were retained. Phase distribution of 18 clinical trials in post-COVID-19 patients affected by anosmia was analyzed in this report.

**Figure 3 diseases-11-00079-f003:**
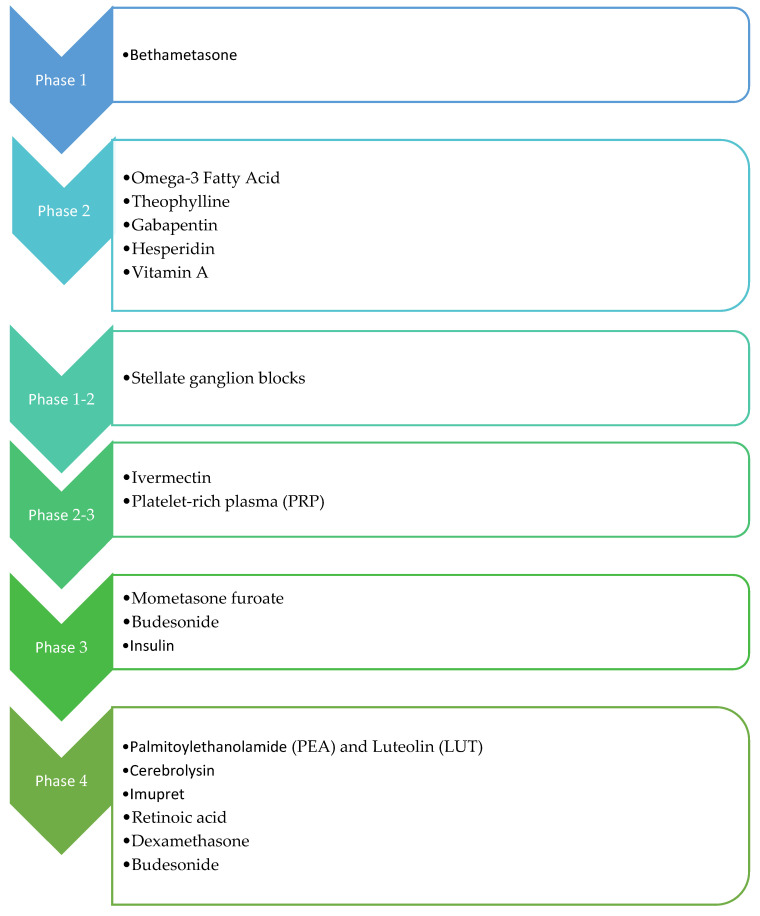
Overview of administered drugs in the clinical trials.

**Figure 4 diseases-11-00079-f004:**
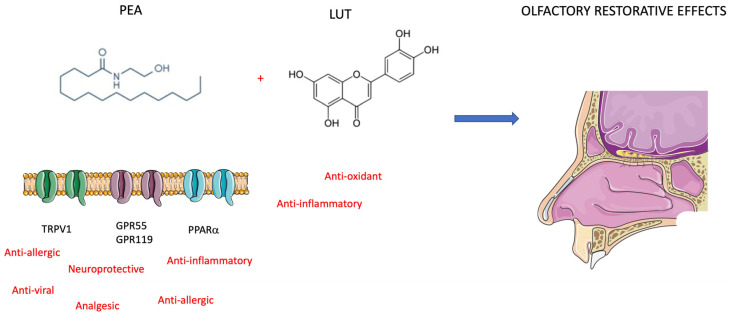
Mechanism of action of PEA and LUT. The effects protective on olfactory epithelium are restorative on post-COVID-19 anosmia (see text for description).

**Figure 5 diseases-11-00079-f005:**
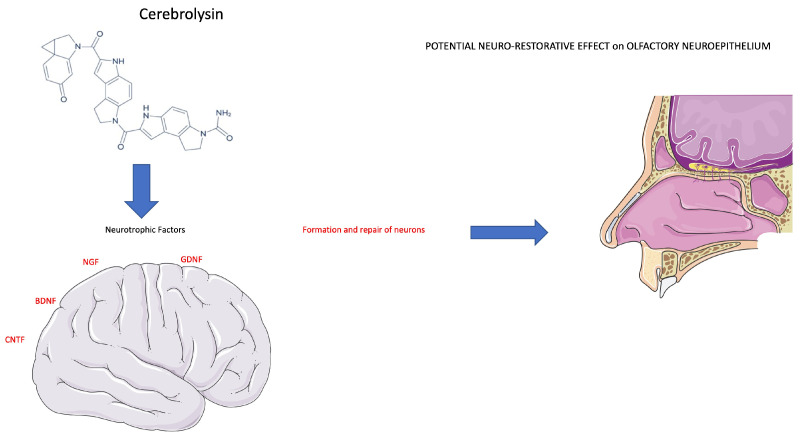
Cerebrolysin stimulates a neurotrophic effect on neuronal cells. The therapeutic effect of Cerebrolysin is attributable to a potential neuro-restorative effect of the olfactory neuroepithelium (see text for description).

**Figure 6 diseases-11-00079-f006:**
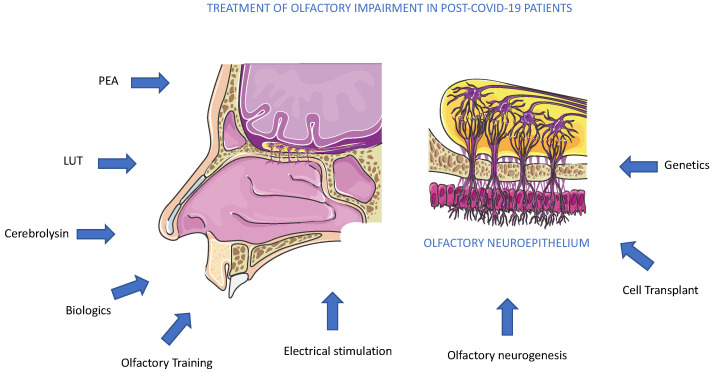
Treatments for olfactory impairment in post-COVID-19 patients. The potential therapeutic approaches, pharmacological, biological, molecular, and genetic, are described in the text. To date, all methods suggested are under experimental trials.

**Table 1 diseases-11-00079-t001:** Summary of protocols analyzed in post-COVID-19 patients affected by anosmia.

Study Title	Intervention/Drug	Phase	Location	Age	Intervention Model and Allocation	Masking
Investigating the Potential Role of Aerosolized Retinoic Acid, a Potent Vitamin A Metabolite, for Treating COVID-19 Anosmia and Retinoic Acid Insufficiency. A Novel Approach for Regaining Sense of Smell. (NCT05002530) [14].	Aerosolized 13-cis retinoic acid plus Vitamin D/Aerosolized All trans-retinoic acid plus Vitamin D	Phase 4	Quan Liu Foshan, Guangdong, China Tamer Haydara Kafr Ash Shaykh, Kafr Elshiekh, Egypt Ministry of Health, Saudi Arabia.	18 years to 70 years	Parallel Assignment Randomized	None (Open Label)
Cerebrolysin for Treatment of COVID-related Anosmia and Ageusia NCT04830943 [15].	Cerebrolysin	Phase 4	Assiut University Hospitals, Faculty of Medicine Assiut, Egypt	20 years to 50 years	Parallel Assignment Randomized	None (Open Label)
COVID-19 Anosmia Study NCT04495816 [13].	Omega-3 Fatty Acid Supplement	Phase 2	Mount Sinai Hospital New York, New York, United States	18 years and older	Parallel Assignment Randomized	Double (Participant, Investigator)
Corticosteroid Nasal Spray in COVID-19 Anosmia NCT04484493 [16].	Mometasone furoate nasal spray	Phase 3	Benha University Hospital, Faculty of Medicine Banhā, Qalubia, Egypt	18 years and older	Parallel Assignment Randomized	None (Open Label)
Stellate Ganglion Block for COVID-19-Induced Olfactory Dysfunction NCT05445921 [17].	Stellate Ganglion Block	Phase 1 Phase 2	Washington University School of Medicine/Barnes Jewish Hospital Saint Louis, Missouri, United States	18 years to 70 years	Single Group Assignment	None (Open Label)
Role of Ivermectin Nanosuspension as Nasal Spray in Treatment of Persistent Post COVID19 Anosmia NCT04951362 [18].	Intranasal spray ivermectin	Phase 2 Phase 3	Zaky Aref Qina, Egypt	18 years to 70 years	Parallel Assignment Randomized	None (Open Label)
Efficacy of Gabapentin for Post-COVID-19 Olfactory Dysfunction NCT05184192 [19,20].	Gabapentin gelatine capsules 300 mg	Phase 2	Washington University Saint Louis, Missouri, United States	18 years to 65 years	Parallel Assignment Randomized	Triple (Participant, Investigator, Outcomes Assessor)
Smell in COVID-19 and Efficacy of Nasal Theophylline NCT04789499 [19].	Theophylline Powder	Phase 2	Washington University School of Medicine in Saint Louis Saint Louis, Missouri, United States	18 years to 70 years	Parallel Assignment Randomized	Double (Participant, Investigator)
Intranasal Insulin for COVID-19-related Smell Loss NCT05461365 [21].	Insulin	Phase 3	Universidad Panamerican Mexico City, Mexico	18 years to 59 years	Single Assignment	None (Open Label)
Anosmia and/or Ageusia and Early Corticosteroid Use NCT04528329 [22].	Early Dexamethasone/Late dexamethasone	Phase 4	Asalam Maadi, Cairo, Egypt	18 years and older	Parallel Assignment Randomized	None (Open Label)
Study of Hesperidin Therapy on COVID-19 Symptoms (HESPERIDIN) NCT04715932 [23].	Hesperidin	Phase 2	Montreal Heart Institute Montréal, Quebec, Canada	18 years and older	Parallel Assignment Randomized	Quadruple (Participant, Care Provider, Investigator, Outcomes Assessor)
Effect of Nasal Steroid in the Treatment of Anosmia Due to COVID-19 Disease NCT04569825 [24].	Ophtamesone	Phase 1	Raid Muhmid Al-Ani Ramadi, Anbar, Iraq	18 years and older	Parallel Assignment Randomized	Double (Participant, Investigator)
Effectiveness and Safety of Platelet Rich Plasma (PRP) on Persistent Olfactory Dysfunction Related to COVID-19 NCT05226546 [25].	Platelet-rich plasma (PRP)	Phase 2 Phase 3	CHU Saint Pierre Bruxelles, Belgium	18 years and older	Parallel Assignment Non-Randomized	None (Open Label)
BNO 1030 Extract (Imupret) in the Treatment of Mild Forms of COVID-19 NCT04797936 [26].	BNO 1030	Phase 4	Ivano-Frankivsk National Medical University Ivano-Frankivsk, Ukraine	18 years to 70 years	Parallel Assignment Randomized	None (Open Label)
Olfactory Disfunction and Co-ultraPEALut NCT04853836 [16].	PEA-LUT	Phase 4	Multicentric Roma, Italy	18 years to 85 years	Parallel Assignment Randomized	Double (Investigator, Outcomes Assessor)
Anosmia Rehabilitation in Patients Post Coronavirus Disease (COVID-19) NCT04374474 Withdrawn (Study withdrawn before any enrollment (site’s research goals adjustments)	Corticosteroid nasal irrigation	Phase 4	St. Joseph’s Health Care London, Ontario, Canada	18 years and older	Parallel Assignment Randomized	None (Open Label)
Olfactory and Neurosensory Rehabilitation in COVID-19-related Olfactory Dysfunction NCT04900415 [27]	Vitamin A	Phase 2	Pamela Youde Nethersole Eastern Hospital Hong Kong, The University of Hong Kong, Queen Mary Hospital, Hong Kong	18 years and older	Parallel Assignment Randomized	Double (Investigator, Outcomes Assessor)

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
