# Peer review of "Post-COVID-19 Anosmia and Therapies: Stay Tuned for New Drugs to Sniff Out"

_diseases, 2023, doi:10.3390/diseases11020079_

Round 1

Reviewer 1 Report (Previous Reviewer 2)

The authors revised the manuscript on some of my points. However, there are many careless mistakes and unchanged or unfixed descriptions. Especially, the revised reference #40 did not assigned at the right position (should be in L.305, not in L.315 for the report of the clinical trial for early corticosteroid use) or some parts of Table 1 would be wrong. How can we ensure the reliability of the present descriptions in the revised manuscript? The authors should very carefully revise the entire manuscript and the references as the last chance.

In addition, I have realized that two my suggestions in Introduction were insufficient. Please consider corrections of L.58 “and the influx of sodium and calcium ions, creating an action potential” -> “for the influx of sodium and calcium ions and subsequent calcium-activated efflux of chloride ions at the olfactory cilia, creating a receptor potential for triggering action potentials at the soma”; L.61 “;” -> “,”.

Moreover, as requested previously, please add the reference # of the report of the clinical trial and (NCT#) at the bottom of study title in Table 1 and for the better readability the clinical trials would be sorted in the increasing order for phase 1 to 4. The 2nd last clinical trial (NCT04374474) was “withdrawn” (shown in an appropriate position).

Regarding unchanged or unfixed cases, please correct the following points:

L.66 “Figure 2” -> “Figure 1”;

L.68 middle “mitral cells” -> “mitral and tufted cells”;

L.68 end “mitral cells” -> “mitral and tufted cells”;

L.82 “Sniffin’ Sticks test battery” -> “Sniffin' Sticks test battery (threshold, discrimination, and identification (TDI) score)”;

L.157 “post-covid-19” -> “post-COVID-19”;

L.168 “Covid-19- associated” -> “COVID-19-associated”;

L.179 “Covid-19” -> “COVID-19”;

L.181 “was executed” -> “Hesperidin therapy was executed as”? (I am not sure. Please check it); L.207–208 “Were injected 1% lidocaine or 1% mepivacaine near the ganglion stellate, after ultrasound guidance identification” -> “After ultrasound guidance identification, 1% lidocaine or 1% mepivacaine was injected near the ganglion stellate”;

L.210, no description of results -> please add them or explain the reason for the no results;

L.215 “post-Covid-19” -> “post-COVID-19”;

L.215–217 “administered them to the patient of one group ivermectin nanosuspension nasal spray and to the patients of group saline nasal spray” -> “administered ivermectin nanosuspension and saline nasal sprays to the treatment and placebo groups, respectively”;

L.220–221 “Olfactory dysfunction treated with PRP increased the olfactory threshold one month after the injection.” -> “At one month after the PRP injection in the olfactory cleft, the mean TDI scores were significantly improved by 6.7 points (from 21.3 ± 7.4 to 28.0 ± 5.0), in contrast to 0.5 points (from 24.5 ± 7.4 to 25.0 ± 7.7) in control group.” (They looked something strange);

L.224–231 should be deleted because of the duplicated description or non-reference description;

L.233 “From Benha University comes the observation to evaluate” -> “The clinical trial at Benha University evaluated” (I am not sure if the authors intended what mean in the original text);

L.242–244 “evaluates the therapy with corticosteroid nasal administration [27]. Was evaluated the efficacy of local budesonide in the management of persistent hyposmia in COVID-19 patients.” -> “evaluated the therapy with corticosteroid nasal administration in the efficacy of local budesonide in the management of persistent hyposmia in COVID-19 patients [27].”;

L.249, please insert the paragraph break.;

L.252–253 “the Threshold, Discrimination, and Identification (TDI) score evaluated with Sniffin' Sticks” -> “TDI scores by Sniffin' Sticks”;

L.265 “Luteolin” -> “Luteolin (LUT)”;

L.274, please insert the paragraph break.;

L.289, please insert the paragraph break.;

L.296, “a day for 14 days” or “for 14 days”? (the former is correct?);

L.300, please insert the paragraph break.;

L.305 “anosmia” -> “anosmia [40]”;

L.313, please insert the paragraph break.;

L.315–319, please reconsider how to describe the withdrawn clinical trial and correct the reference (the present one is for 13 cis retinoic acid);

L.342, L.343, L.350, L.356, L.359, as requested previously, please add the reference# as evidence;

L.345, please correct the reference# (32?, the present one is for STRA6);

L.365, please enlarge the text size such as TRPV1, Anti-allergic, and so on;

L.369 “post-covid-19” -> “post-COVID-19”;

L.370 & L.374 “Luteolin” -> “LUT”;

L.377 “[43,44]” -> “[47,48]”;

L.379 “45” -> “36”;

L.406, please show the reference# for each of drugs with reconsideration for the description of the withdrawn clinical trial;

L.412 “Retinoic Acid aerosolized ” -> “aerosolized retinoic acid” (as the same as in L.304);

L.412–415 “the authors suggest the loss of stra6 function through blocking it by COVID-19 spike protein which binds to it with high affinity as a result it may hijack the signalling pathway leading to retinoic acid synthesis disruption and vitamin A deficiency. This is because” -> “Elkazzaz et al. indicated that the administrated retinoic acid would facilitate to cellular uptake of retinol in immune cells through the impaired transport function of STRA6 receptor by binding to the RBD of SARS-CoV-2 spike protein leading to improving the immune system homeostasis [40, 51+]. This may also be applicable to the olfactory epithelial homeostasis, because” (51+: Elkazzaz M, et al. STRA6 (vitamin A receptor), as a Novel binding receptor of COVID-19 (A breakthrough). Research Square (preprint) https://doi.org/10.21203/rs.3.rs-892203/v3 (2021). available from https://www.researchsquare.com/article/rs-892203/v3) (hijack and retinoic acid synthesis may be not related in this case, meaning that they should be deleted for maintaining a logical flow);

L.418, please insert the paragraph break.;

L.436 “post-covid” -> “post-COVID-19”;

Please consider a proper English editing service for better readability of the manuscript.

Author Response

Dear Editor,

We thank the editor for his/her letter and the reviewers for their comments on our manuscript (Manuscript ID: diseases-2413587). Those comments are all valuable and very helpful for revising and improving our paper, as well as the important guiding significance to our research. We have studied the comments carefully and have made corrections which we hope meet with approval. Revised portions are marked in yellow on the paper. The Table 1 was completed as requested. The references increased from 60 to 79. Please see below for point-by-point responses.  The main corrections in the manuscript and the response to the reviewer's comments are as following:

Reply to the Reviewer # 1 comments:
Point 1: The authors revised the manuscript on some of my points. However, there are many careless mistakes and unchanged or unfixed descriptions. Especially, the revised reference #40 did not assigned at the right position (should be in L.305, not in L.315 for the report of the clinical trial for early corticosteroid use) or some parts of Table 1 would be wrong. How can we ensure the reliability of the present descriptions in the revised manuscript? The authors should very carefully revise the entire manuscript and the references as the last chance.
Response 1: We are very thankful for reviewer’s nice advice. We have revised the entire manuscript. All points were addressed. The references were all revised and we added other 19 references. Several parts were discussed and Table and Figures were improved. Reference #40 (#48) was shifted to L.305 (L.317) as suggested.

Point 2: In addition, I have realized that two my suggestions in Introduction were insufficient. Please consider corrections of L.58 “and the influx of sodium and calcium ions, creating an action potential” -> “for the influx of sodium and calcium ions and subsequent calcium-activated efflux of chloride ions at the olfactory cilia, creating a receptor potential for triggering action potentials at the soma”; L.61 “;” -> “,”.

Response 2: As suggested, sentence L.58 was replaced; the typo L.61 was corrected.

Regarding unchanged or unfixed cases, please correct the following points:

Point 3: L.66 “Figure 2” -> “Figure 1”;

Response 3: corrected;

Point 4: L.68 middle “mitral cells” -> “mitral and tufted cells”;

Response 4: replaced;

Point 5: L.68 end “mitral cells” -> “mitral and tufted cells”;

Response 5:  replaced;

Point 6: L.82 “Sniffin’ Sticks test battery” -> “Sniffin' Sticks test battery (threshold, discrimination, and identification (TDI) score)”;

Response 6: added

Point 7: L.157 “post-covid-19” -> “post-COVID-19”;

Response 7: corrected;

Point 8: L.168 “Covid-19- associated” -> “COVID-19-associated”;

Response 8: corrected;

Point 9: L.179 “Covid-19” -> “COVID-19”;

Response 9: corrected;

Point 10: L.181 “was executed” -> “Hesperidin therapy was executed as”? (I am not sure. Please check it);

Response 10: the sentence was replaced with:“executed a trial with the aim to determine the effects of short-term treatment with hesperidin, a medicinal plant with antiviral action, on COVID-19 symptoms, was completed. The 216 participants were divided into two groups: the active comparator and the placebo comparator. Patients received investigational drug study medication (2 capsules of 500mg of Hesperidin) at the same time every day at the same time in the evening, at bedtime with water.”

Point11: L.207–208 “Were injected 1% lidocaine or 1% mepivacaine near the ganglion stellate, after ultrasound guidance identification” -> “After ultrasound guidance identification, 1% lidocaine or 1% mepivacaine was injected near the ganglion stellate”;

Response 11: the sentence was replaced.

Point 12: L.210, no description of results -> please add them or explain the reason for the no results;

Response 12: the sentence was explained: ”The recruitment status was completed but no study results were posted on ClinicalTrials.gov”

Point 13: L.215 “post-Covid-19” -> “post-COVID-19”;

Response 13: corrected;

Point 14: L.215–217 “administered them to the patient of one group ivermectin nanosuspension nasal spray and to the patients of group saline nasal spray” -> “administered ivermectin nanosuspension and saline nasal sprays to the treatment and placebo groups, respectively”;

Response 14: replaced;

Point 15: L.220–221 “Olfactory dysfunction treated with PRP increased the olfactory threshold one month after the injection.” -> “At one month after the PRP injection in the olfactory cleft, the mean TDI scores were significantly improved by 6.7 points (from 21.3 ± 7.4 to 28.0 ± 5.0), in contrast to 0.5 points (from 24.5 ± 7.4 to 25.0 ± 7.7) in control group.” (They looked something strange);

Response 15: replaced;

Point 16: L.224–231 should be deleted because of the duplicated description or non-reference description;

Response 16: the sentences were deleted;

Point 17: L.233 “From Benha University comes the observation to evaluate” -> “The clinical trial at Benha University evaluated” (I am not sure if the authors intended what mean in the original text);

Response 17: the sentence was replaced;

Point 18: L.242–244 “evaluates the therapy with corticosteroid nasal administration [27]. Was evaluated the efficacy of local budesonide in the management of persistent hyposmia in COVID-19 patients.” -> “evaluated the therapy with corticosteroid nasal administration in the efficacy of local budesonide in the management of persistent hyposmia in COVID-19 patients [27].”;

Response 18: the sentence was replaced;

Point 19: L.249, please insert the paragraph break.;

Response 19: added

Point 20: L.252–253 “the Threshold, Discrimination, and Identification (TDI) score evaluated with Sniffin' Sticks” -> “TDI scores by Sniffin' Sticks”;

Response 20: the sentence was replaced;

Point 21: L.265 “Luteolin” -> “Luteolin (LUT)”;

Response 21: replaced

Point 22: L.274, please insert the paragraph break.;

Response 22: inserted

Point 23: L.289, please insert the paragraph break.;

Response 23: inserted

Point 24: L.296, “a day for 14 days” or “for 14 days”? (the former is correct?);

Response 24: for 14 days is correct.

Point 25: L.300, please insert the paragraph break.;

Response 25: inserted

Point 26: L.305 “anosmia” -> “anosmia [40]”;

Response 26: replaced

Point 27: L.313, please insert the paragraph break.;

Response 27: inserted

Point 28: L.315–319, please reconsider how to describe the withdrawn clinical trial and correct the reference (the present one is for 13 cis retinoic acid);

Response 28: the sentence was entirely rewritten:”They evaluated the administration of dexamethasone 6 mg/day per os and 4 mg/day intravenously in two groups of 15 patients with community-acquired pneumonia. The study aimed to use dexamethasone as early as laboratory evidence of high inflammatory markers compared to the late dexamethasone administration upon the deterioration of cases. The time frame of 6 weeks to recover from anosmia has been evaluated, and the study is running.” (Ref.22.                  Spoorenberg SM, Deneer VH, Grutters JC, Pulles AE, Voorn GP, Rijkers GT, Bos WJ, van de Garde EM. Pharmacokinetics of oral vs. intravenous dexamethasone in patients hospitalized with community-acquired pneumonia. Br J Clin Pharma-col. 2014 Jul;78(1):78-83. doi: 10.1111/bcp.12295.)

Point 29: L.342, L.343, L.350, L.356, L.359, as requested previously, please add the reference# as evidence;

Response 29: several new references were added as follow:

  1. Rankin L, Fowler CJ. The Basal Pharmacology of Palmitoylethanolamide. Int J Mol Sci. (2020) 21:7942–62. doi: 10.3390/ijms21217942
  2. Jaggar SI, Hasnie FS, Sellaturay S, Rice AS. The anti-hyperalgesic actions of the cannabinoid anandamide and the putative CB2 receptor agonist palmitoylethanolamide in visceral and somatic inflammatory pain. Pain.(1998) 76:189–99. doi: 10.1016/S0304-3959(98)00041-4
  3. Lo Verme J, Fu J, Astarita G, La Rana G, Russo R, Calignano A, et al. The nuclear receptor peroxisome proliferator-activated receptor-alpha mediates the anti-inflammatory actions of palmitoylethanolamide. Mol Pharmacol.(2005) 67:15–9. doi: 10.1124/mol.104.006353
  4. Solorzano C, Zhu C, Battista N, Astarita G, Lodola A, Rivara S, et al. Selective N-acylethanolamine-hydrolyzing acid amidase inhibition reveals a key role for endogenous palmitoylethanolamide in inflammation. Proc Natl Acad Sci U S A.(2009) 106:20966–71. doi: 10.1073/pnas.0907417106
  5. Paterniti I, Cordaro M, Campolo M, Siracusa R, Cornelius C, Navarra M, et al. Neuroprotection by association of palmitoylethanolamide with luteolin in experimental Alzheimer's disease models: the control of neuroinflammation. CNS Neurol Disord Drug Targets.(2014) 13:1530–41. doi: 10.2174/1871527313666140806124322

Point 30: L.345, please correct the reference# (32?, the present one is for STRA6);

Response 30: the ref. 52 was replaced.

Point 31: L.365, please enlarge the text size such as TRPV1, Anti-allergic, and so on;

Response 31: the fig. text was enlarged;

Point 32: L.369 “post-covid-19” -> “post-COVID-19”;

Response 32: replaced;

Point 33: L.370 & L.374 “Luteolin” -> “LUT”;

Response 33: replaced;

Point 34: L.377 “[43,44]” -> “[47,48]”;

Response 34: replaced;

Point 35: L.379 “45” -> “36”;

Response 35: replaced;

Point 36: L.406, please show the reference# for each of drugs with reconsideration for the description of the withdrawn clinical trial;

Response 36: all references were added [12,13,48,26];

Point 37: L.412 “Retinoic Acid aerosolized ” -> “aerosolized retinoic acid” (as the same as in L.304); ”;

Response 37: replaced;

Point 38: L.412–415 “the authors suggest the loss of stra6 function through blocking it by COVID-19 spike protein which binds to it with high affinity as a result it may hijack the signalling pathway leading to retinoic acid synthesis disruption and vitamin A deficiency. This is because” -> “Elkazzaz et al. indicated that the administrated retinoic acid would facilitate to cellular uptake of retinol in immune cells through the impaired transport function of STRA6 receptor by binding to the RBD of SARS-CoV-2 spike protein leading to improving the immune system homeostasis [40, 51+]. This may also be applicable to the olfactory epithelial homeostasis, because” (51+: Elkazzaz M, et al. STRA6 (vitamin A receptor), as a Novel binding receptor of COVID-19 (A breakthrough). Research Square (preprint) https://doi.org/10.21203/rs.3.rs-892203/v3 (2021). available from https://www.researchsquare.com/article/rs-892203/v3) (hijack and retinoic acid synthesis may be not related in this case, meaning that they should be deleted for maintaining a logical flow);

Response 38: the full period and reference were added;

Point 39: L.418, please insert the paragraph break.;

Response 39: inserted;

Point 40: L.436 “post-covid” -> “post-COVID-19”;

Response 40: replaced;

Thanks again for the reviewer's comments, we revised each quoted sentence again.

We appreciate for Editors/Reviewers’ warm work earnestly and really hope that our modification of this paper can get your precious recognition, which is of great significance to us.

Reviewer 2 Report (New Reviewer)

In the review, titled "Post-Covid-19 Anosmia and Therapies: Stay tuned for new drugs to sniff out", the authors very well explained the importance of clinical trials specifically targeting towards the COVID-19 anosmics. 

In section 3.3, phase 1-2 clinical trials authors mentioned 1 clinical trails in phase 1-2, however, no data on significant changes in the olfactory function was mentioned. This questions the effectiveness of the treatment.

The authors did not address treatment effectiveness in long COVID anosmics and related clinical trials.

In the discussion, authors write "we discussed all therapeutics employed in several anosmia clinical trials and atleast only PEA and cerebrolysin represent promising molecules acting to reduced olfactory impairment", I would suggest not to over exaggerate the conclusion.

No further changes required.

Author Response

Dear Editor,

We thank the editor for his/her letter and the reviewers for their comments on our manuscript (Manuscript ID: diseases-2413587). Those comments are all valuable and very helpful for revising and improving our paper, as well as the important guiding significance to our research. We have studied the comments carefully and have made corrections which we hope meet with approval. Revised portions are marked in yellow on the paper. The Table 1 was completed as requested. The references increased from 60 to 79. Please see below for point-by-point responses.  The main corrections in the manuscript and the response to the reviewer's comments are as following :

Reply to the Reviewer # 2 comments:

Point 1: In the review, titled "Post-Covid-19 Anosmia and Therapies: Stay tuned for new drugs to sniff out", the authors very well explained the importance of clinical trials specifically targeting towards the COVID-19 anosmics. 

Response 1: Thank you for the reviewer important suggestions and comments. We have revised the entire manuscript. All points were addressed.

Point 2: In section 3.3, phase 1-2 clinical trials authors mentioned 1 clinical trails in phase 1-2, however, no data on significant changes in the olfactory function was mentioned. This questions the effectiveness of the treatment.

Response 2: The study was updated with the last post deposited. The question was explained by the following sentence Lines 214-215: “The recruitment status was completed but no study results were posted on ClinicalTrials.gov.”

Point 3: The authors did not address treatment effectiveness in long COVID anosmics and related clinical trials.

Response 3: We discussed this issue and added these sentences to Discussion section: (lines 470-479)

In addition, regarding the treatment effectiveness in long COVID anosmics, the clinical experience is limited. This is due mainly both the research on human olfactory dysfunctions it is developing, as the investigations on the olfactive trans-duction mechanism, the cellular and molecular profile of neuroepithelial cells, cognitive and memory elaboration of olfactory area, and too little time have passed since the end of the pandemic to already have data available on what is defined as a post-covid syndrome [4]. More recently, an international consensus on the management of post-covid-19 olfactory dysfunction, suggested that olfactory training remains the recommended management protocol, and treatment with systemic corticosteroids is not recommended until new studies are completed [73]. (Wu TJ, Yu AC, Lee JT. Management of post-COVID-19 olfactory dysfunction. Curr Treat Options Allergy. 2022;9(1):1-18. doi: 10.1007/s40521-021-00297-9.)

Point 4: In the discussion, authors write "we discussed all therapeutics employed in several anosmia clinical trials and atleast only PEA and cerebrolysin represent promising molecules acting to reduced olfactory impairment", I would suggest not to over exaggerate the conclusion.

Response 4: The sentence was rewritten and replaced with (Lines 480-482): “We discussed several therapeutics employed in several anosmia clinical trials, and among active trials, PEA and Cerebrolysin represented promising molecules acting to reduce olfactory impairment”

Point 5: Comments on the Quality of English Language

No further changes required.

Response 5: Thanks again for the reviewer's comments, we revised each quoted sentence again.

We appreciate for Editors/Reviewers’ warm work earnestly and really hope that our modification of this paper can get your precious recognition, which is of great significance to us.

Round 2

Reviewer 1 Report (Previous Reviewer 2)

The revised manuscript well addressed the pointed issues except for the necessity of deletion of the sentence in L.252-253.

Some of the reference# in the text were wrong (31 (L.170) -> 30; 32 (L.172) -> 31; 17 (L.206) -> 34. Please recheck if all the others are correct.

Author Response

Dear Editor,

We thank the editor for his/her letter and the reviewers for their comments on our manuscript (Manuscript ID: diseases-2413587).

Reply to the Reviewer # 1 comments (II round):

Point 1: The revised manuscript well addressed the pointed issues except for the necessity of deletion of the sentence in L.252-253.

Response 1: sentence L.252-253 was deleted.

Point 2: Some of the reference# in the text was wrong (31 (L.170) -> 30; 32 (L.172) -> 31; 17 (L.206) -> 34. Please recheck if all the others are correct.

Response 2: The references were corrected and all were rechecked.

We are very thankful for the reviewer’s nice advice. We have revised the entire manuscript. All points were addressed. (Manuscript ID: diseases-2413587)

This manuscript is a resubmission of an earlier submission. The following is a list of the peer review reports and author responses from that submission.

Round 1

Reviewer 1 Report

The article “Anosmia and Therapies: Stay tuned for new drugs to sniff out” presents a review on olfactory related studies regarding COVID-19. While the framework for a review is presented, detailed analysis and further ideas are yet to be discussed in the text. This reviewer suggests major revisions before this manuscript can be accepted, see the following comments.

1.       The authors should clearly describe what the number of phases refer to in the form of a diagram. This will benefit all readers to clearly gauge the research topic regardless of scientific background.

2.       Figure 1 should present additional information. Refer to Comment 1. In its current form, Figure 1 is not so informative.

3.       Are studies able to be classified by other key markers such as “region”, “population” etc., this will make the review more detailed compared to its current form.

4.       The authors should double check the terminology. It may be better to write “Phase X Clinical trials”.

5.       The reviewer highly recommends a thorough English proofreading, whilst most of the manuscript is fine, there are still many errors that appear throughout the text.

6.       Line 209 “South University” Which University and Country is this.

7.       Figure 3 doesn’t relay any essential information. The authors should make this Figure much more informative than the current version.

8.       Figure 4 doesn’t relay any essential information. The authors should make this Figure much more informative than the current version.

9.       Technical comparison between different olfactory methods used for testing should also be analysed in terms of sensitivity, reliability, selectivity etc. There was a lot of discussion about false-positive tests regarding COVID-19 which means sensor analysis or machine analysis is also a necessary section for this review.

10.   More technical discussion is required, at the current point it is only summarising the work of others and not providing a more detailed discussion or interpretation of the overall results.

11.   The review also needs to include the others own thoughts as to how to proceed this field further in the future. Discussion should contain mechanisms linking the inhibition of olfactory receptors and COVID-19 etc. Moreover, a final figure summarising the authors future and perspectives into olfactory diagnosis must be included.

Author Response

Dear Editor,

We are appreciative of the comments and suggestions made by reviewers and believe that addressing their comments and concerns substantially improved our manuscript. All the changes in the manuscripts have been evidenced (yellow). Please see below for point-by-point responses.  As suggested, the manuscript was checked by a native English-speaking colleague.

We are looking forward to hearing from you soon.

Comments and Suggestions for Authors

The article “Anosmia and Therapies: Stay tuned for new drugs to sniff out” presents a review on olfactory related studies regarding COVID-19. While the framework for a review is presented, detailed analysis and further ideas are yet to be discussed in the text. This reviewer suggests major revisions before this manuscript can be accepted, see the following comments.

  1. The authors should clearly describe what the number of phases refer to in the form of a diagram. This will benefit all readers to clearly gauge the research topic regardless of scientific background.

R1 We added a new figure (Fig.2) in the form of diagram.

  1. Figure 1 should present additional information. Refer to Comment 1. In its current form, Figure 1 is not so informative.

R2 Figure 1 was changed in Fig.2.

  1. Are studies able to be classified by other key markers such as “region”, “population” etc., this will make the review more detailed compared to its current form.

R3 The studies summarized in Table 1 were classified as suggested. A new version of Table 1 was added.

  1. The authors should double check the terminology. It may be better to write “Phase X Clinical trials”.

R4 The terminology was checked, and the suggestion was added.

  1. The reviewer highly recommends thorough English proofreading, whilst most of the manuscript is fine, there are still many errors that appear throughout the text.

R5 Thorough English proofreading were performed

  1. Line 209 “South University” Which University and Country is this.

R6 The sentence “ In Egypt, South Valley University…….” was added.

  1. Figure 3 doesn’t relay any essential information. The authors should make this Figure much more informative than the current version.

R7 Fig.3 was eliminated and substituted with a new Figure, named Fig.4

  1. Figure 4 doesn’t relay any essential information. The authors should make this Figure much more informative than the current version.

R8 Fig.4 was eliminated and substituted with a new Figure, named Fig.5

  1. Technical comparison between different olfactory methods used for testing should also be analysed in terms of sensitivity, reliability, selectivity etc. There was a lot of discussion about false-positive tests regarding COVID-19 which means sensor analysis or machine analysis is also a necessary section for this review.

R9 The manuscript focuses on clinical trials, many reports are ongoing or paused, and only a few studies have been concluded. The technical comparison of olfactory methods creates confusion between studies from different countries. This topic is beyond the scope of this work, and we are aware that an effort by the medical and research community should focus on standardizing the technical aspects of smell testing. We will be happy to perform in a subsequent study the discussion about false-positive tests regarding COVID-19, such as sensor analysis or machine.

  1. More technical discussion is required, at the current point it is only summarising the work of others and not providing a more detailed discussion or interpretation of the overall results.

R10 The discussion was reorganized and rewrote aimed to provide a more detailed results interpretation.

  1. The review also needs to include the others own thoughts as to how to proceed this field further in the future. Discussion should contain mechanisms linking the inhibition of olfactory receptors and COVID-19 etc. Moreover, a final figure summarising the authors future and perspectives into olfactory diagnosis must be included.

R11 The discussion was reorganized and rewrote aimed to provide a more detailed results interpretation. A final figure elucidating future perspectives on olfactory impairment treatment was added (Fig.6)

Thank you for appreciating our work and for helping us to make it clearer to the reader with these suggestions.

We hope that you find the revised version of the article to merit its publication in the Special Issue " Translational Neurobiology: Molecular, Cellular, and Sensory Neuroscience in Human Health and Diseases”.

Kind regards,

Christian Barbato M.D. PhD,

National Research Council (CNR)

Institute of Biochemistry and Cell Biology (IBBC)

Department of Sense Organs,

 University Sapienza of Rome,

Viale del Policlinico, 155

00161 Rome, Italy

Reviewer 2 Report

The authors reviewed clinical trials of therapeutic drugs for anosmia due to SARS-CoV-2 in the database in the clinicaltrials.gov. The clinical trials were well collected. However, the readability of the manuscript is poor especially in the description of the olfactory system in Introduction. The parts mentioned below should be rewritten. In addition, in most of the clinical trials reviewed, the authors did not describe the reported results (improved or not significant?), which would be very important information for the readers. The authors should describe them in a contrasted manner in appropriately divided paragraphs. Moreover, by a quick PubMed search by the keywords of “COVID-19, anosmia, clinical trial”, we can easily find some interesting reports. Although I am not sure whether the reports met the criterion for this survey or not, the addition of the following two papers (no mentioned phases) would make this review article more informative. Generally, more clearly structured (and appropriately positioning of necessary information), concise and contrasted descriptions for benefits and risks of the drugs for anosmia therapy would improve the readability of this manuscript.

1)    As a clinical trial, “Hasanpour M, et al. Efficacy of Covexir® (Ferula foetida oleo-gum) treatment in symptomatic improvement of patients with mild to moderate COVID-19: A randomized, double-blind, placebo-controlled trial. Phytother. Res. 36:4504-4515 (2022). doi: 10.1002/ptr.7567. Epub 2022 Jul 27” would be interesting for the readers as an oral medicine for anosmia/hyposmia therapy, although I am not sure whether these authors’ cases classified as moderate anosmia were anosmia or hyposmia. It is likely that non-hospitalized patients with severe anosmia got better after taking drug twice daily for more than three days.

2)    Intranasally applied insulin fast-dissolving film improved the olfactory detection scores and olfactory discrimination values in patients with anosmia (Mohamad SA, Badawi AM, Mansour HF. Insulin fast-dissolving film for intranasal delivery via olfactory region, a promising approach for the treatment of anosmia in COVID-19 patients: Design, in-vitro characterization and clinical evaluation. Int J Pharm. 601:120600 (2021). doi: 10.1016/j.ijpharm.2021.120600).

3)    Please start the sections of 3.1–3.6 with their outlines for better readability.

4)    I would recommend the authors to ask an English editing service. Please improve the readability by dividing the section into appropriate paragraphs and omitting detailed typical methods such as “in a randomized control trial” (L.145–146); 

5)    “were randomized into a control group and treatment, receiving” (L.147) -> “received”; 

6)    “, which is a 12-item instrument” (L.150) -> deleted; “where ,,, performance,” (L.151) -> deleted; 

7)    “University of ,,, Test (“ & “)” (L.162, not the first appearance) -> deleted; 

8)    “The test ,,, 0-14” (L.173–175) -> deleted; 

9)    “0-4” -> “0–4” and similar errors for numeric ranges; 

10) L.178 & L.200–201 & L.205 & L.237 look something wrong in the syntax; 

11) L.163 & L.177 & L.194 & L.207 & L.215 & L.224 lacked the reported results (improved or not significant?);

12) Please describe the reported evidence for the pharmacological effects of insulin as a drug for anosmia in L.245;

13) Please add the significant therapeutic effects of the drugs on molecular or cellular levels or olfactory functional scores or no reported evaluation at the present statuses (ongoing? and associated evidence in animal models, if possible) and their references in Table 1; 

14) “after three administrations of” (L.245) and “perfomed three times” would be duplicated. Please merge them. In addition, please describe the results of the odor threshold and discrimination scores in L.252. 

15) Please unify the terminology such as Sars-Cov2 (L.19, L.22, L.33, legend in Fig.1, etc) -> SARS-CoV-2; 

16) COVID (L.245) and Covid-19 (L.165, L.257, legend in Fig.1) -> COVID-19; 

17) Luteolin (L.261, not the first appearance) -> LUT; 

18) post-covid-19 (L.153) & post-COVID (L.267) -> post-COVID-19; 

19) hair cells (L.53, for hearing sense) -> ciliated cells; 

20) olfactory sensory neuronal cells (L.56, unusual) -> olfactory sensory neurons (OSNs); 

21) microvillar -> microvillar cells; 

22) basal -> basal cells; 

23) olfactory gland -> Bowman's glands; 

24) Olfactory neurons (L.57) -> OSNs; 

25) odor detection and induce activation (L.58) -> detection of odorant molecules by odorant binding-induced activation; 

26) chloride channels and the efflux of chloride ions (L.60) -> cyclic nucleotide-gated channels and the influx of sodium and calcium ions; 

27) nasal cavity through their dendrites (L.62) -> nasal lamina propria and the cribriform plate; 

28) “unique odor receptor that projects to the glomeruli and synapses with cells (L.63)” -> “single type of olfactory receptors; the signals of which are convergently transmitted to one or two glomeruli and thereby connecting tufted and mitral cells”; 

29) slit (L.72) -> cleft; 

30) UPSIT (L.76, spelling out on the first appearance) -> University of Pennsylvania Smell Identification Test (UPSIT); 

31) Sniffing Sticks (L.76, L.247) -> Sniffin' Sticks; 

32) L.270–272 -> “Cerebrolysin is a mixture of porcine-derived neuropeptides and free amino acids including nerve growth factor (NGF), brain-derived neurotrophic factor (BDNF), ciliary neurotrophic factor (CNTF), enkephalins, orexin, and P21 [34,35], approved for use as treatment for dementia [35], stroke [34,36], cognitive impairment [34?] and traumatic brain injury (TBI) [37].” as an introduction of Cerebrolysin. The references 43–46 would be inserted after the reference 33 and the previous references 34–42 would be corrected as the references 38–46;  

33) x (L.288) -> ×;

34) STRA 6 (L.293) -> STRA6 (L.291); 

35) olfactory receptors and olfactory sensory cells and neurons (L.293–294) -> OSNs expressing olfactory receptors;

36) use. [36] (L.305) -> use [40]. (by the inserted references 43–46 at 34 as described above);

37) Please adjust the size of Figure 2. In addition, please describe the summary of Figure 2 with “(Figure 2)” in the text.

38) Please add the reference # in L.321, L.324, L.328, L.330, L.332, L.333, L.340, L.343, L.346, L348–349 for each disease. 

39) palmitoylethanolamide (PEA) (L.324–325) -> PEA;

40) Vanilloid receptor transient potential 1 (TRPV1) (in Figure 3) -> transient receptor potential vanilloid 1 (TRPV1) channel. Potential pathways from these pharmacological effects to therapeutic effects on improvement of anosmia would be added in Figure 3 for better readability;

41) In L.330–339, please clearly describe the activations or inhibitions of all four channels/receptors including TRPV1 in the text as well as in Figure 3. Please appropriately insert “(Figure 3)” in the text.

42) Luteolin (L.366, L.368, L.370) -> LUT;

43) L.371–376 -> “In addition to the treatment for dementia, acute stroke, cognitive impairment and TBI described above,” after modifying and moving to P.9 as described above;

44) Please appropriately insert “(Figure 4)” in the text. What does the brown item mean? It did not look any neuropeptides. Please reconstruct the figure in an at-a-glance easy understanding way for potential effects on the improvement of anosmia via the most likely pathways.

45) Please correct many syntactic errors and style errors such as “days” (L.50) -> “days,”, “-14” (L.288) -> “for 14”, periods before [xx] and so on.

Author Response

Dear Editor,

We are appreciative of the comments and suggestions made by reviewers and believe that addressing their comments and concerns substantially improved our manuscript. All the changes in the manuscripts have been evidenced (yellow). Please see below for point-by-point responses.  As suggested, the manuscript was checked by a native English-speaking colleague.

We are looking forward to hearing from you soon.

Comments and Suggestions for Authors

The authors reviewed clinical trials of therapeutic drugs for anosmia due to SARS-CoV-2 in the database in the clinicaltrials.gov. The clinical trials were well collected. However, the readability of the manuscript is poor especially in the description of the olfactory system in Introduction. The parts mentioned below should be rewritten. In addition, in most of the clinical trials reviewed, the authors did not describe the reported results (improved or not significant?), which would be very important information for the readers. The authors should describe them in a contrasted manner in appropriately divided paragraphs. Moreover, by a quick PubMed search by the keywords of “COVID-19, anosmia, clinical trial”, we can easily find some interesting reports. Although I am not sure whether the reports met the criterion for this survey or not, the addition of the following two papers (no mentioned phases) would make this review article more informative. Generally, more clearly structured (and appropriately positioning of necessary information), concise and contrasted descriptions for benefits and risks of the drugs for anosmia therapy would improve the readability of this manuscript.

1)    As a clinical trial, “Hasanpour M, et al. Efficacy of Covexir® (Ferula foetida oleo-gum) treatment in symptomatic improvement of patients with mild to moderate COVID-19: A randomized, double-blind, placebo-controlled trial. Phytother. Res. 36:4504-4515 (2022). doi: 10.1002/ptr.7567. Epub 2022 Jul 27” would be interesting for the readers as an oral medicine for anosmia/hyposmia therapy, although I am not sure whether these authors’ cases classified as moderate anosmia were anosmia or hyposmia. It is likely that non-hospitalized patients with severe anosmia got better after taking drug twice daily for more than three days.

R1) We thank the reviewer for this suggestion, but the study did not fit the inclusion criteria shown in this manuscript.

2)    Intranasally applied insulin fast-dissolving film improved the olfactory detection scores and olfactory discrimination values in patients with anosmia (Mohamad SA, Badawi AM, Mansour HF. Insulin fast-dissolving film for intranasal delivery via olfactory region, a promising approach for the treatment of anosmia in COVID-19 patients: Design, in-vitro characterization and clinical evaluation. Int J Pharm. 601:120600 (2021). doi: 10.1016/j.ijpharm.2021.120600).

R2) We thank the reviewer for this suggestion, and the study was included in the manuscript (Ref. 29) 

3)    Please start the sections of 3.1–3.6 with their outlines for better readability.

R3) We thank the reviewer for this suggestion on sections 3.1-3.6 

4)    I would recommend the authors to ask an English editing service. Please improve the readability by dividing the section into appropriate paragraphs and omitting detailed typical methods such as “in a randomized control trial” (L.145–146); 

R4) The manuscript was read by a native English speaker. The sentence was replaced.  

5)    “were randomized into a control group and treatment, receiving” (L.147) -> “received”; 

R5) The sentence was deleted and replaced. 

6)    “, which is a 12-item instrument” (L.150) -> deleted; “where ,,, performance,” (L.151) -> deleted; 

R6) The terms were deleted.

7)    “University of ,,, Test (“ & “)” (L.162, not the first appearance) -> deleted; 

R7) The terms were deleted. 

8)    “The test ,,, 0-14” (L.173–175) -> deleted; 

 R8) The terms were deleted.

9)    “0-4” -> “0–4” and similar errors for numeric ranges; 

 R9) The error was corrected.  

10) L.178 & L.200–201 & L.205 & L.237 look something wrong in the syntax; 

R10) The sentences were rewritten.

11) L.163 & L.177 & L.194 & L.207 & L.215 & L.224 lacked the reported results (improved or not significant?); rispondere al punto

R11) The results were added, if reported in the database.

12) Please describe the reported evidence for the pharmacological effects of insulin as a drug for anosmia in L.245;

R12) The reported evidence were described.

13) Please add the significant therapeutic effects of the drugs on molecular or cellular levels or olfactory functional scores or no reported evaluation at the present statuses (ongoing? and associated evidence in animal models, if possible) and their references in Table 1; 

R13) With the aim to disclose the significant therapeutic effects, the discussion was reorganized and rewrote; two new figures, Fig.4 and Fig, 5, were added; Table 1 was replaced with a new version. 

14) “after three administrations of” (L.245) and “perfomed three times” would be duplicated. Please merge them. In addition, please describe the results of the odor threshold and discrimination scores in L.252. 

R14) The terms were merged. The discrimination scores were added.

15) Please unify the terminology such as Sars-Cov2 (L.19, L.22, L.33, legend in Fig.1, etc) -> SARS-CoV-2; 

R15) The terminology was unified.

16) COVID (L.245) and Covid-19 (L.165, L.257, legend in Fig.1) -> COVID-19; 

R16) The terminology was unified.  

17) Luteolin (L.261, not the first appearance) -> LUT; 

R17) Luteolin was replaced with LUT

18) post-covid-19 (L.153) & post-COVID (L.267) -> post-COVID-19; 

 R18) The terminology was unified.  

19) hair cells (L.53, for hearing sense) -> ciliated cells; 

R19) The ciliated cells were replaced instead hair cells.   

20) olfactory sensory neuronal cells (L.56, unusual) -> olfactory sensory neurons (OSNs); 

R20) The term olfactory sensory neurons (OSNs) was added.

21) microvillar -> microvillar cells; 

R21) The term microvillar cells was added.

22) basal -> basal cells; 

R22) The term basal cells; was added.

23) olfactory gland -> Bowman's glands; 

R23) The term Bowman's glands was added.

24) Olfactory neurons (L.57) -> OSNs; 

R24) The term OSNs was added.

25) odor detection and induce activation (L.58) -> detection of odorant molecules by odorant binding-induced activation; 

R25) The sentence was replaced.

26) chloride channels and the efflux of chloride ions (L.60) -> cyclic nucleotide-gated channels and the influx of sodium and calcium ions; 

R26) The sentence was replaced.

27) nasal cavity through their dendrites (L.62) -> nasal lamina propria and the cribriform plate; 

 R27) The sentence was replaced.

28) “unique odor receptor that projects to the glomeruli and synapses with cells (L.63)” -> “single type of olfactory receptors; the signals of which are convergently transmitted to one or two glomeruli and thereby connecting tufted and mitral cells”; 

R28) The sentence was replaced.

29) slit (L.72) -> cleft; 

R29) The word was replaced.

30) UPSIT (L.76, spelling out on the first appearance) -> University of Pennsylvania Smell Identification Test (UPSIT); 

 R30) The acronymous was replaced.

31) Sniffing Sticks (L.76, L.247) -> Sniffin' Sticks; 

 R31) The term was corrected.

32) L.270–272 -> “Cerebrolysin is a mixture of porcine-derived neuropeptides and free amino acids including nerve growth factor (NGF), brain-derived neurotrophic factor (BDNF), ciliary neurotrophic factor (CNTF), enkephalins, orexin, and P21 [34,35], approved for use as treatment for dementia [35], stroke [34,36], cognitive impairment [34?] and traumatic brain injury (TBI) [37].” as an introduction of Cerebrolysin. The references 43–46 would be inserted after the reference 33 and the previous references 34–42 would be corrected as the references 38–46;  

 R32) The suggested reorganization of the sentences was replaced, and the references were renumbered.

33) x (L.288) -> ×;

 R33) The symbol was corrected. 

34) STRA 6 (L.293) -> STRA6 (L.291); 

 R34) The acronymous was replaced.

35) olfactory receptors and olfactory sensory cells and neurons (L.293–294) -> OSNs expressing olfactory receptors;

R35) The sentence was replaced.

36) use. [36] (L.305) -> use [40]. (by the inserted references 43–46 at 34 as described above);

R36) The references were renumbered.

37) Please adjust the size of Figure 2. In addition, please describe the summary of Figure 2 with “(Figure 2)” in the text.

R37) Figure 2 was reorganized and named Fig. 3, and reported in the text. 

38) Please add the reference # in L.321, L.324, L.328, L.330, L.332, L.333, L.340, L.343, L.346, L348–349 for each disease. 

R38) The reference n.42 (Clayton, P.; Hill, M.; Bogoda, N.; Subah, S.; Venkatesh, R. Palmitoylethanolamide: A Natural Compound for Health Management. Int. J. Mol. Sci. 202122, 5305. doi.org/10.3390/ijms22105305) summarize the references for each disease.

39) palmitoylethanolamide (PEA) (L.324–325) -> PEA;

R39) The acronymous was replaced. 

40) Vanilloid receptor transient potential 1 (TRPV1) (in Figure 3) -> transient receptor potential vanilloid 1 (TRPV1) channel. Potential pathways from these pharmacological effects to therapeutic effects on improvement of anosmia would be added in Figure 3 for better readability;

R40) Fig.3 was reorganized and added as Fig.4  

41) In L.330–339, please clearly describe the activations or inhibitions of all four channels/receptors including TRPV1 in the text as well as in Figure 3. Please appropriately insert “(Figure 3)” in the text.

R41) The insert Fig.4 was added. 

42) Luteolin (L.366, L.368, L.370) -> LUT;

 R42) The term was replaced with acronymous. 

43) L.371–376 -> “In addition to the treatment for dementia, acute stroke, cognitive impairment and TBI described above,” after modifying and moving to P.9 as described above;

R43) The sentence was modified.  

44) Please appropriately insert “(Figure 4)” in the text. What does the brown item mean? It did not look any neuropeptides. Please reconstruct the figure in an at-a-glance easy understanding way for potential effects on the improvement of anosmia via the most likely pathways.

R44) Fig.4 was reorganized and added as Fig.5  

45) Please correct many syntactic errors and style errors such as “days” (L.50) -> “days,”, “-14” (L.288) -> “for 14”, periods before [xx] and so on.

R45) Syntax and typos were corrected in all parts of the manuscript.  

Thank you for appreciating our work and for helping us to make it clearer to the reader with these suggestions.

We hope that you find the revised version of the article to merit its publication in the Special Issue " Translational Neurobiology: Molecular, Cellular, and Sensory Neuroscience in Human Health and Diseases”.

Kind regards,

Christian Barbato M.D. PhD,

National Research Council (CNR)

Institute of Biochemistry and Cell Biology (IBBC)

Department of Sense Organs,

 University Sapienza of Rome,

Viale del Policlinico, 155

00161 Rome, Italy

Round 2

Reviewer 1 Report

The article “Anosmia and Therapies: Stay tuned for new drugs to sniff out” describes some phase studies about covid-19’s influence on the human olfactory bulb.

Whilst this reviewer offered comments to the authors previous revision through major revision, it appears not to have been considered seriously. For the new manuscript, the figures are misleading, have small text, no logical meaning etc. Fig. 6 has red underlining of many words. The schematics appear to have very little meaning and vast parts of the text have nothing to do with olfactory sensing. Overall, the structure of the manuscript is not suitable as a review article. This reviewer suggests the manuscript be rejected.